# Metabolomic and Transcriptomic Changes Underlying the Effects of *L*-Citrulline Supplementation on Ram Semen Quality

**DOI:** 10.3390/ani13020217

**Published:** 2023-01-06

**Authors:** Guodong Zhao, Xi Zhao, Jiachen Bai, Airixiati Dilixiati, Yukun Song, Aerman Haire, Shangshang Zhao, Aikebaier Aihemaiti, Xiangwei Fu, Abulizi Wusiman

**Affiliations:** 1Laboratory of Animal Genetic Breeding & Reproduction, Xinjiang Agricultural University, Urumqi 830052, China; 2State Key Laboratory of Sheep Genetic Improvement and Healthy Breeding, Institute of Animal Husbandry and Veterinary Sciences, Xinjiang Academy of Agricultural and Reclamation Sciences, Shihezi 832000, China

**Keywords:** metabolome, transcriptome, *L*-Cit, semen quality regulation, rams

## Abstract

**Simple Summary:**

Our previous study found that the addition of *L*-citrulline (*L*-Cit) to feed improved ram semen density as well as vitality and sperm mitochondrial membrane potential. In this experiment, after feeding 12 g/d of *L*-Cit to rams for 70 consecutive days, the contents of amino acids, sugars, and pyruvate in seminal plasma, in addition to the changes in metabolites in plasma, were analyzed by metabolomics, and the differentially expressed genes in the hypothalamus and testes were further studied through a transcriptome analysis. The results showed that *L*-Cit could increase the contents of pyruvate and most amino acids in seminal plasma, and that the differentially expressed genes in the hypothalamus and testes were significantly enriched in protein-metabolism-related pathways. This shows that *L*-Cit can improve the contents of amino acids in rams’ semen by promoting the protein metabolism of the hypothalamus–testes axis, thus improving semen quality.

**Abstract:**

This study examined the effects of *L*-Cit supplementation on ram semen quality through metabolomics and transcriptomics. A total of 16 rams were randomly categorized into two groups. The control group was fed a basic diet, whereas the experimental group received feed supplemented with 12 g/d of *L*-Cit. Semen and blood were collected from the rams on days 0 and 72 to measure sugar, pyruvate, amino acid, and nontargeted metabolite contents. Additionally, hypothalamic and testicular tissues were collected for a transcriptomic analysis. We found 27 differential metabolites between the control and experimental groups, of which 21 were downregulated (*p* < 0.05) and 6 were upregulated (*p* < 0.05). Compared with the control group, xylose and pyruvate contents in seminal plasma increased by 43.86% and 162.71%, respectively (*p* < 0.01). Additionally, the levels of 11 amino acids showed a significant increase in seminal plasma (*p* < 0.01). Furthermore, 961 and 715 differentially expressed genes were detected in the hypothalamic and testicular tissues, respectively. The pathways of significant enrichment in the hypothalamus and testes were protein digestion, absorption, glycolysis/gluconeogenesis, and amino as well as nucleotide sugar metabolisms. In the present study, *L*-Cit improved protein synthesis and blood metabolism, consequently increasing the contents of most amino acids in ram seminal plasma. Specifically, the hypothalamus controlled the expression of glycolysis/gluconeogenesis-related genes in the testes through its metabolites released into the serum, thereby providing energy for sperm production, which led to a decrease in the sugar content of seminal plasma.

## 1. Introduction

In our previous study, we demonstrated that supplementing sheep with 12 g/d of *L*-citrulline (*L*-Cit) improved their sperm motility, density, and mitochondrial membrane potential. Additionally, the levels of reproductive hormones and oxidation resistance in the serum and seminal plasma significantly increased; however, the mechanisms underlying these effects remain unclear [1]. Therefore, it is crucial to analyze the above phenomena and explore the mechanism by which *L*-Cit improves semen quality. Seminal plasma, an essential component of semen, is mainly secreted by the accessory sex glands. It contains amino acids, minerals, sugars, and hormones, which provide sperm with energy, nutrition, and protection. Seminal plasma also plays a crucial role in male fertility and sperm function [2]. Certain seminal plasma proteins are involved in sperm capacitation and immune response regulation in the female reproductive tract [3], including fertilization in the oviduct [4,5]. Importantly, seminal plasma can reduce inflammatory reactions after artificial insemination [6]. Based on these characteristics, seminal plasma can serve as an evaluation index and predictor for semen quality and male fertility [7]. The seminal plasma of patients with oligospermia and asthenospermia has also been examined using proton nuclear magnetic resonance, revealing increased levels of glycerophosphate choline, citric acid, histidine, *L*-phenylalanine acid, and alanine after receiving *Tribulus terrestris* L. supplementation, which indicates that amino acid contents in seminal plasma influence semen quality. Notably, sugar and glucose, present in seminal plasma and semen diluent, respectively, comprise the most common energy sources for sperm. Furthermore, D-fructose has been reported as an energy source present in semen, as it can be directly metabolized by sperm [8]. Hence, the metabolites in seminal plasma are critical for sperm energy, motility, pH maintenance, and metabolic activity regulation [9]. They also influence downstream gene and protein expression [10].

*L*-Cit is a nonprotein amino acid, and although not involved in protein synthesis in vivo, it is involved in post-translational modifications [11,12]. *L*-Cit does not degrade in the rumen of sheep and cattle [13,14]; instead, it is absorbed into the portal vein through intestinal epithelial cells, subsequently reaching the liver. Under the action of succinic acid, argininosuccinatelyase, and arginase, citrulline–arginine–ornithine and urea cycles are activated. Nitric oxide (NO) metabolism, occurring in macrophages via arginase and induced NO synthase, is involved in reproduction at all levels, from the brain to peripheral organs; hence, it is also known as “sexual gas.” In the male reproductive system, NO is vital in the functions of the testis, epididymis, and vas deferens [15]. Research regarding *L*-Cit primarily focuses on disease prevention and treatment; however, its effects on reproductive performance have rarely been investigated. Consequently, the mechanism underlying the effects of *L*-Cit on ram semen quality remains unclear. Therefore, this study investigated the mechanism by which *L*-Cit regulates ram semen quality by detecting changes in the amino acid and sugar contents of seminal plasma, plasma metabolites, and gene expression in ram hypothalamic and testicular tissues.

## 2. Materials and Methods

### 2.1. Study Conditions

This study was conducted at the Tuokesun Huishang Ecological Animal Husbandry Co., Ltd., Turpan, China. (87°14′05″–89°11′08″ E, 41°21′14″–43°18′11″ N) for 72 days from May 2020 to November 2020. The average outdoor and indoor temperature values were 15.1 °C and 17.1 °C, respectively.

### 2.2. Reagents

Chromatographic-grade acetonitrile and methanol were purchased from the American Sigma Corporation. Formic acid and standard samples were purchased from Aladdin Biochemical Technology Co., Ltd., Shanghai, China. Liquid injection bottles and ultrapure water were obtained from Agilent Technologies. Other reagents included aceQ universal SYBR qPCR Master Mix Q511-02 (Nuovizan Biotechnology Co., Ltd., Nanjing, China), TRIpure Reagent RN01, RNasin RNA inhibitor RN3501 (Adlai Biotechnology Co., Ltd., Beijing, China), Revertaid reverse transcriptase EP0442, dNTPmix R0191, DNase I EN0521 (Thermos Fisher), chloroform (Tianli Chemical Reagent Co., Ltd., Tianjin, China), isopropanol, and absolute ethanol (Sinopharm Chemical Reagent Co., Ltd., Shanghai, China).

A centrifuge (TGL-18M) from Kaida Scientific Instrument Co., Ltd., Hunan, China. A real-time PCR instrument (StepOnePlus), and a 96-well PCR plate were purchased from Applied Biosystems Co., Ltd., Waltham, MA, USA. An RNase-free gun head, an RNase-free centrifuge tube, and an RNase-free octet PCR tube were purchased from Extragene Co., Ltd. Guangzhou, China.

### 2.3. Study Design

Overall, 16 healthy 1-year-old male Turpan black rams with an average weight of 61.92 ± 4.87 kg were selected for semen collection. Following semen collection using an artificial vagina, they were randomly categorized into control and test groups, each including eight rams. The control group was fed a basic diet, while the experimental group was supplemented with 12 g/d of *L*-Cit. The spermatogenesis cycle in sheep lasts for approximately 55 days. The supplemental feeding period was 72 days. The study protocol was approved by the Ethics Committee of the Xinjiang Agricultural University (protocol permit number: 2020032, 7 May 2020).

### 2.4. Feed Management

All rams were provided with food and water ad libitum. They were fed using a total mixed ration feeder truck at 10:30 and 16:30 daily. As a supplement, 12 g of *L*-Cit was mixed with 100 mL of tap water in a clean bottle and fully dissolved. Rams were restrained at 10:00 am every day; after using the left thumb to move the tongue to the corner of the mouth, the rams were fed within 1 min, with the *L*-Cit solution poured into the other corner of the mouth. After swallowing, they were released. All of the animals were fed under the same conditions. The roughage, concentrate composition, and nutritional level in each experimental group were similar. All of the animals were allowed to drink water and move freely. Table 1 presents the detailed diet compositions and nutritional levels of the rams.

### 2.5. Sample Collection and Testing

Semen was collected between 1 (recorded as day 0) and 71 days before the test and morning feeding. After collection, the samples were immediately transported to the laboratory. Next, two semen samples from the same ram group were mixed and centrifuged at 3500 rpm for 15 min. The supernatant was extracted from the four samples (*n* = 4) using a pipette gun, transferred into a 1.5 mL cryotube, and stored in liquid nitrogen until further testing to determine the sugar, pyruvic acid, and amino acid contents in seminal plasma. After semen collection from all of the rams, 5 mL of blood was collected before feeding from the jugular vein using a disposable needle. Two blood samples from the same ram group were mixed and centrifuged at 3500 rpm for 15 min. Next, the supernatant was extracted from the four samples using a pipette gun, transferred into a 1.5 mL cryotube, and stored in liquid nitrogen for further testing to determine nontarget sera metabolites.

On the 72nd day, three random rams from each group (the control and experimental groups) were slaughtered. The rams’ skulls were opened, and the hypothalamic tissue was collected with tweezers and stored in a 2 mL cryopreservation tube. Furthermore, the right testicle was excised, and testicular tissue samples were collected as 2 × 2 cm sections, after which they were stored in a 2 mL cryopreservation tube. Finally, the tissues were stored in liquid nitrogen.

### 2.6. Serum Sample Preparation

The 12 ram serum samples (*n* = 4) were thawed in an ice bath, and 100 μL–1.5 mL was added to centrifuge tubes containing a methanol–acetonitrile (1:1) mixed solution of 400 μL. Next, the samples were mixed thoroughly via a vortex for 30 s and ultrasonically mixed for 10 min. The samples were then placed in a refrigerator at −20 °C for 1 h and centrifuged at 4 °C at 13,000 rpm for 15 min; 300 μL of supernatant was collected and dried in a vacuum freeze dryer. After thoroughly mixing the supernatant with an acetonitrile and water (1:1) solution for 30 s, the sample was centrifuged for 10 min at 4 °C at 13,000 rpm; 50 μL of the supernatant was transferred into a sample tube.

### 2.7. Preparation of Samples for Sugar and Amino Acid Determination in Seminal Plasma

After thawing the samples (*n* = 4) in an ice bath, 0.1 mL of the samples was added to 700 μL of an 80% methanol solution. The sample was incubated at 4 °C for 60 min and centrifuged at 4 °C and 12,000 rpm for 10 min. Finally, the supernatant was used for a liquid chromatography–mass spectrometry (LC–MS/MS) analysis (Allwegene Technology Co., Ltd., Beijin, China)

After thawing the samples in an ice bath, 50 μL of the samples was added to a 1.5 mL centrifuge tube with 450 μL of precooled methanol (containing a 100 ng/mL internal standard) solution. After being shaken for 1 min, the sample was incubated at 4 °C for 60 min and centrifuged at 4 °C at 12,000 rpm for 10 min. The supernatant was finally used for an LC–MS/MS analysis.

### 2.8. RNA Extraction

Hypothalamic (*n* = 3) and testicular tissue (*n* = 3) samples frozen in liquid nitrogen were added to liquid nitrogen in a sterilized mortar for thorough grinding. After grinding, 1 mL of TRIzol reagent was added for digestion, followed by 200 µL of trichloromethane. After mixing, it was allowed to stand for 10 min and centrifuged at 13,000 rpm at 4 °C for 15 min. Next, the supernatant was transferred into a sterilized Eppendorf tube containing the same amount of isopropanol precooled at 4 °C, thoroughly mixed, allowed to stand at room temperature for 10 min, and finally centrifuged at 13,000 rpm at 4 °C for 10 min. The residue in the supernatant tube, which was the extracted RNA, was discarded. Afterward, 1 mL of 75% ethanol was added into the Eppendorf tube and centrifuged at 12,000 rpm and 4 °C for 15 min. The supernatant was discarded, dried, and the appropriate amount of deionized water was added for dissolution. Subsequently, 1.5% agarose gel electrophoresis was conducted to detect the RNA extraction quality. The bands were clear, and no fracture or towing was observed after imaging (Figure 1).

### 2.9. RNA Sample

The average total RNA integrity of the hypothalamic tissues in the control group was 8.23 (7.6–9.3), and that in the test group was 8.3 (7.1–8.9). However, the total RNA integrity of testicular tissues in the control and test group was 8.83 (8.7–8.9) and 8.77 (8.7–8.9), respectively. Because the average RNA integrity of the control and test groups was >8, it met the cDNA library building standard and was used for follow-up tests.

### 2.10. Sample Testing

#### 2.10.1. Liquid Phase and Mass Spectrometric Conditions of Serum Samples

Column: Waters ACQUITY UPLC BEH Amide 1.7 µm, 2.1 × 100 mm.

Mobile phase: Phase A was ultrapure water (containing 25 mm of ammonium acetate and 25 mm of ammonia). Phase B was acetonitrile: flow rate: 0.5 mL/min; column temperature: 40 °C; and injection volume: 2 μL, with a liquid-phase elution gradient.

Electrospray ion source temperature: 650 °C; mass spectrum voltage: 5500 V (positive ion), −4500 V (negative ion); decluster voltage: 60 V; ion source gas: gas 1, 60 psi, gas 2, 60 psi; and gas curtain gas: 30 psi. Impact-induced ionization parameters were high.

#### 2.10.2. Liquid Phase and Mass Spectrometric Conditions for Sugar Content Determination in Seminal Plasma

Chromatographic separation condition: column temperature: 40 °C; flow rate: 0.55 mL/min. The mobile phase comprised (A) water and (B) acetonitrile. The running time was 10 min, and the injection volume was 6 µL.

The mass spectrometry conditions were as follows: ESI source: 35 arb gas curtain, 7 arb collision gas, 4500 V ion spray voltage, 450 °C ion source temperature, and 55 arb (GAS1 and GAS2) ion source gas.

#### 2.10.3. Liquid Phase and Mass Spectrometric Conditions for Amino Acid Content Determination in Seminal Plasma

Chromatographic separation conditions: column temperature: 40 °C; flow rate: 0.45 mL/min. The mobile phase comprised (A) water (0.1% formic acid) and (B) acetonitrile (0.1% formic acid). Additionally, the running time was 12 min and the injection volume was 6 µL.

The mass spectrometry conditions were as follows: ESI source: 35 arb gas curtain, 7 arb collision gas, 4500 V ion spray voltage, 450 °C ion source temperature, and 55 arb (GAS1 and GAS2) ion source gas.

### 2.11. RNA Library Construction and Quality Inspection

An NEBNext^®^ Ultra™ RNA Library Prep Kit for Illumina^®^ Kit was used for developing the library. The cDNA of approximately 250–300 bps was screened with an AMPure XP beads kit. After PCR amplification, the product was purified to obtain the final library. Following library construction, the Qubi2.0 accounting protein quantizer was used for preliminary quantification. The library was diluted to 1.5 ng/µL, and an Agilent 2100 biological analyzer was used to detect the library insertion size. Next, qRT-PCR was performed to accurately determine the effective library concentration (>2 nm) to ensure library quality. Furthermore, after database construction, different libraries were combined based on the requirements of effective concentration and target off-line data volume. Subsequently, Illumina sequencing was performed to obtain the sequence information of the fragments to be evaluated.

### 2.12. Data Quality Control

Fastq 0.20.1 software was employed to filter the original data. Illumina Casava version 1.8 and FastQC version 0.11.9 software were used to detect the error rate and GC content of high-throughput sequencing data. HISAT version 2.2.1 software was used to map the transcriptome sequencing reads of clean data to obtain a reference group of sheep genes (oar-v 4.0).

### 2.13. Data Processing

#### 2.13.1. Statistical Analysis

Serum samples were analyzed via MasterView (SCIEX ZenoTOF™ 7600, USA) software and a metabonomic analysis platform. The results are shown in Table 2.

#### 2.13.2. Metabolomic Data Preprocessing

Values of <50% and <80% in quality management (QC) and real samples, respectively, were retained in the serum samples, and a single peak was filtered. The total ion current of each sample was normalized, and the data were standardized. The minimum value half method was used to fill in the missing values in the original data. All QC samples were filtered by a coefficient of variation (CV) of >30%; ions with a CV of >30% fluctuated greatly during the test, and hence were not included in the downstream statistical analysis.

After integrating sugars, amino acids, and pyruvate in seminal plasma using 

MultiQuant software, the seminal syrup contents were calculated using the standard curve and internal standard point method. The data obtained were preliminarily sorted using MS Excel 2010. SAS 9.0 statistical software was used to conduct multiple comparisons with least-square means. A *p*-value of <0.05 was considered significant, and <0.01 indicated a highly significant difference.

#### 2.13.3. Transcriptome Data Processing

Comparison of transcriptome sequencing data and sheep genome in the NCBI database (https://ftp.ncbi.nlm.nih.gov/genomes/refseq/vertebrate_mammalian/Ovis_aries/latest_assembly_versions/GCF_002742125.1_Oar_ramboµillet_v1.0/ accessed on 19 January 2022). The PANTHER HMM scoring tool and library version 16.0 were downloaded from PANTHER (ftp://ftp.pantherdb.org/hmm_scoring/current_release accessed on 19 January 2022). The HMM alignment algorithm of the PANTHER HMM scoring tool was used to align the protein sequences of differentially expressed genes with the sequences in PANTHER HMM library version 16.0 to obtain the GO classification ID in the PANTHER database. The results were then analyzed by PANTHER Generic Mapping (http://pantherdb.org/webservices/go/overrep.jsp accessed on 19 January 2022). A KEGG enrichment analysis was performed by uploading the protein ID corresponding to the differentially expressed gene to KOBAS (http://kobas.cbi.pku.edu.cn/genelist/ accessed on 19 January 2022). A genetic interaction analysis was performed at https://www.string-db.org/ accessed on 19 January 2022.

## 3. Results

### 3.1. Hierarchical Cluster Analysis

The first principal component score of the serum samples in each group was significant; all of the samples were within a 95% confidence interval (Figure 2A). The level of cluster analysis of the serum samples on days 0 and 72 in the test and control groups was clear (Figure 2B). A total of 170 metabolites were annotated. In the test group, 163 and 73 differential metabolites were annotated on days 0 and 72, respectively, and in the control group 133 and 6 differential metabolites were annotated on days 0 and 72, respectively. On day 72, 132 metabolites in the control and test groups, 19 differential metabolites, and 105 specific metabolites in the control group were annotated (Figure 2C).

### 3.2. Screening Map of the Differential Metabolites

On days 0 and 72, 342 differential metabolites were detected in the ram serum samples in the control group; 112 differential metabolites showed a significant change, of which 44 were downregulated (*p* < 0.05), 68 were upregulated (*p* < 0.05), and the remaining 230 exhibited no significant effect (Figure 3A).

A total of 342 differential metabolites were detected in the ram serum samples in the test group on days 0 and 72; 124 differential metabolites showed significant changes, of which 63 differential metabolites were downregulated (*p* < 0.05), 61 were upregulated (*p* < 0.05), and the remaining 218 demonstrated no significant effect (Figure 3B).

After *L*-Cit supplementation for 72 days, a total of 342 differential metabolites were detected in in the ram serum samples in the control and experimental groups; 27 differential metabolites showed a significant difference, of which 21 were downregulated (*p* < 0.05), 6 were upregulated (*p* < 0.05), and the remaining 315 were not significantly different (Figure 3C).

### 3.3. KEGG Analysis of Differential Metabolites among the Combinations

#### 3.3.1. KEGG Analysis of Serum Metabolism in the Groups on Days 0 and 72

According to the KEGG analysis of the blood metabolites in both of the groups on days 0 and 72, there were three most significant pathways among the top 20 main enriched pathways in the serum, including pyrimidine as well as phenylalanine metabolism and arginine biosynthesis (Figure 4).

The abscissa represents the impact of each path, and the ordinate represents the path name. The results of the metabolic pathway analysis are shown in the bubble chart. The bubble color represents the *p*-value of the enrichment analysis; the redder the color, the more significant the enrichment. The point size represents the number of enriched differential metabolites. A: day 0; B: day 72.

#### 3.3.2. KEGG Analysis of Differential Serum Metabolism in L-Cit-Supplemented Rams on Days 0 and 72

According to the KEGG analysis of the blood metabolites in *L*-Cit-supplemented rams on days 0 and 72, the top 20 main enrichment pathways in the serum included the seven most significant pathways: pyrimidine metabolism; protein digestion and absorption; mineral absorption; arginine biosynthesis; arginine and proline metabolism; aminoacyl tRNA biosynthesis; and alanine, aspartate, and glutamate metabolism (Figure 5).

The abscissa represents the impact of each path, and the ordinate represents the path name. The results of the metabolic pathway analysis are shown in the bubble chart. The bubble color represents the *p*-value of the enrichment analysis; the redder the color, the more significant the enrichment. The point size represents the number of enriched differential metabolites. A: ram serum before *L*-Cit feeding; C: ram serum on day 72.

#### 3.3.3. KEGG Analysis of Differential Serum Metabolism in the Control and Test Groups on Day 72

According to the KEGG analysis of the differential blood metabolites in both of the groups on day 72, the top 20 main enrichment pathways in the serum included the nine most significant metabolic pathways: valine, leucine, and isoleucine biosynthesis; protein digestion and absorption; mineral absorption; central carbon metabolism of cancer; amino acid biosynthesis; arginine and proline metabolism; aminoacyl tRNA biosynthesis; alanine, aspartate, and glutamate metabolism; and ATP-binding cassette (ABC) transport (Figure 6).

The abscissa represents the impact of each path, and the ordinate represents the path name. The results of the metabolic pathway analysis are shown in the bubble chart. The bubble color represents the *p*-value of the enrichment analysis; the redder the color, the more significant the enrichment. The point size represents the number of enriched differential metabolites. B: control group on day 72; C: test group on day 72.

### 3.4. Effect of Supplementary L-Cit Feeding on Sugar Content in Ram Seminal Plasma

Compared with the control group, the xylose and pyruvate contents in the experimental group increased by 43.86% and 162.71%, respectively (*p* < 0.01), Time, intergroup, and interaction showed a higher significant effect (*p* < 0.01). The xylose content in the experimental group was 10.63% higher than that in the control group (*p* < 0.01). There was no significant effect between groups and in the interaction between groups and time (*p* > 0.05); however, time exhibited a highly significant effect (*p* < 0.01). The lactose content increased, but time, intergroup, and time and group interaction had no significant effect (*p* > 0.05).

Compared with the control group, the arabinose and maltose contents in the experimental group decreased by 10.83% (*p* < 0.01) and 10.58% (*p* < 0.01), respectively. There were significant effects among groups, time, and group and time interaction (*p* < 0.01). Furthermore, compared with the control group, the ribose content in the experimental group decreased by 28.10% (*p* < 0.05); there was a significant effect between the groups (*p* < 0.05); and time had a highly significant effect (*p* < 0.01), as did intergroup interaction between groups and time (*p* < 0.05). However, the glucose and fructose contents in the experimental group were 4.13% and 3.55%, respectively, which were lower than those in the control group, but this difference was not significantly different (*p* > 0.05). Time showed a highly significant effect (*p* < 0.01). These results are summarized in Table 3.

### 3.5. Effect of L-Cit Supplementation on Amino Acid Levels in Ram Seminal Plasma

Compared with the control group, the contents of *L*-valine, *L*-threonine, *L*-arginine, *L*-tryptophan, *L*-isoleucine, *L*-lysine, *L*-methionine, *L*-aspartic acid, acetyllysine, sarcosine, and *L*-homoserine increased by 36.20%, 48.21%, 37.21%, 108.14%, 13.01%, 30.7%, 109.75%, 26.27%, 23%, 14.46%, and 48.71%, respectively (*p* < 0.01). S-adenosylmethionine in seminal plasma was 34.08% lower than in the control group (*p* < 0.01). Time, intergroup, and time and intergroup interaction had a highly significant effect (*p* < 0.01).

Compared with the control group, the *L*-proline, *L*-citrulline, and *L*-glutamate contents in ram seminal plasma in the experimental group increased. The glycine content in the experimental group was 23.21% lower than that in the control group. There were significant differences in terms of groups, time, and the interaction between groups and time (*p* < 0.05).

Conversely, compared with the control group, the *L*-leucine content in the experimental group was 12.48% higher (*p* < 0.01). Additionally, time had no significant effect on the content (*p* > 0.05), and the interaction between groups and time also showed a highly significant effect (*p* < 0.01). The dimethylglycine content in ram seminal plasma was 3.02% higher than that in the control group (*p* < 0.01), the time factor had a significant impact (*p* < 0.05), and the interaction between groups and time showed a highly significant effect (*p* < 0.01).

Compared with the control group, *L*-lysine in the experimental group increased by 27.82% (*p* < 0.05). Although time showed no significant effect (*p* > 0.05), the interaction between groups and time was significantly different between the groups (*p* < 0.05). Moreover, *L*-pyroglutamate and *L*-*O*-phosphoserine contents in ram seminal plasma were 24.99% and 32.09%, respectively, higher than those in the control group. The factors between the groups were significantly different (*p* < 0.05), as was the interaction between groups and time, whereas the time showed no significant impact (*p* > 0.05). Furthermore, the *L*-ornithine hydrochloride content in ram seminal plasma was 8.29% higher than that in the control group (*p* < 0.05), the time factor had a highly significant effect (*p* < 0.01), and the interaction between groups and time also showed a significant effect (*p* < 0.05).

Additionally, compared with the control group, *N*-acetyl-glutamate, *L*-alanine, *L*-kynurenine hydrate, and *L*-cystine contents showed an increase in ram seminal plasma in the experimental group. Interestingly, the *L*-phenylalanine content in this group was 14.52% lower than that in the control group. There was no significant effect of intergroup (*p* > 0.05), time (*p* < 0.01), and the interaction between groups and time (*p* > 0.05) in this group.

Furthermore, the *L*-histidine, *L*-serine, *L*-homoproline, *L*-hydroxyproline, 1-methyl-*L*-histidine, *N*-acetyl-glutamate, *S*-adenosyl-*L*-homocysteine, and homocysteine contents in rams’ seminal plasma increased compared with the control group. The *D*-tyrosine content decreased, but there was no significant effect between groups, time, and the interaction between groups and time (*p* > 0.05). These results are summarized in Table 4.

### 3.6. Quality of Hypothalamic and Testicular Tissue Sequencing Data

The Illumina NovaSeq platform was used to sequence a cDNA library of 12 samples, and quality control of the sequencing data was performed. A total of 559,580,148 reads were detected, and 540,529,557 were mapped to the reference genome. The CG content ranged from 48.75% to 53.52% (Table 5).

We found various gene expression patterns in each tissue. The gene expression levels were similar in each group (Figure 7A). Additionally, two groups were categorized according to the mRNA transcriptome profiling of the hypothalamic (Figure 7B) and testicular (Figure 7C) tissues. There were 15,256 genes identified in ram hypothalamic tissues, including 14396 genes in the control and test groups, of which 436 and 424 genes were specifically expressed in the control and test groups, respectively (Figure 7D). Overall, 18,829 genes were identified in ram testicular tissues, including 17,151 genes in the control and test groups, of which 978 and 700 genes were specifically expressed in the control and test groups, respectively (Figure 7E). Furthermore, there were 1619 genes in the ram hypothalamic and testicular tissues, including 57 genes in the hypothalamic and testicular tissues, of which 904 and 658 genes were specifically expressed in the hypothalamus and testis, respectively (Figure 7F). A total of 24,350 differentially expressed genes were detected in ram hypothalamic tissues in both of the groups, including 524 upregulated genes (*p* < 0.05) and 437 downregulated genes (*p* < 0.05; Figure 7G). Overall, 28,654 differentially expressed genes were detected in ram testicular tissue in both of the groups, including 260 upregulated genes (*p* < 0.05) and 455 downregulated genes (*p* < 0.05; Figure 7H).

### 3.7. GO Enrichment Analysis of Differentially Expressed Genes in Hypothalamic Tissues

The most significant 30 terms in hypothalamic tissues were selected according to an increasing *p*-value. Overall, 205 differential genes were classified and enriched for GO function. Collagen trimer (GO: 0005581), dynein complex (GO: 0030286) in the cellular component (CC), microtubule-associated complex (GO: 0005875), and molecular function (MF) containing an extracellular matrix structural component (GO: 0005201) were significantly increased (Figure 8A).

Additionally, the most significant 30 terms from testicular tissues were selected; <30 were supplemented based on an increasing *p*-value. Overall, 210 differential genes were grouped and enriched for GO function. These were the most significantly different terms in biological process (BP): antigen processing and presentation (GO:0019882), immune response (GO:0006955), immune system process (GO:0002376), cell adhesion (GO:0007155), cell adhesion (GO:0022610), homophilic cell adhesion by plasma membrane adhesion molecules (GO:0007156), cell–cell adhesion through plasma membrane adhesion molecules (GO:0098742), cell–cell adhesion (GO:0098609), nucleosome assembly (GO:0006334), and chromatin assembly (GO:0031497). Notably, there were nine significantly different terms in CC: MHC protein complex (GO:0042611), MHC protein complex (GO:0042613), plasma membrane protein complex (GO:0098797), plasma membrane part (GO:0044459), extracellular matrix (GO:0031012), plasma membrane (GO:0005886), cell junction (GO:0030054), and cell periphery (GO:0071944) (Figure 8B).

The abscissa includes the GO term, and the ordinate is the significance level of GO term enrichment, represented by −log10 (Padj); different colors represent different functional classifications.

### 3.8. KEGG Enrichment Analysis of Differentially Expressed Genes in Hypothalamic and Testicular Tissues

According to the KEGG enrichment of ram hypothalamic and testicular tissues, the most significant 20 KEGG pathways were displayed as a scatter diagram (Figure 9A): protein digestion and absorption (oas04974), hypothalamus glycolysis/gluconeogenesis (oas00010), amino sugar and nucleotide sugar metabolism (oas00520), leishmaniasis (oas05140), cell adhesion molecules (oas04514), *Staphylococcus aureus* infection (oas05150), galactose metabolism (oas00052), allograft rejection (oas05330), viral myocarditis (oas05416), type 1 diabetes mellitus (oas04940), autoimmune thyroid disease (oas05320), HIF-1 signaling pathway (oas04066), and human T-cell leukemia virus 1 infection (oas05166) in testicular tissues (Figure 9B).

The abscissa indicates the ratio of the number of differential genes annotated in the KEGG pathway corresponding to that of differential genes; the ordinate indicates the KEGG pathway.

### 3.9. Protein–Protein Interaction in Ram Hypothalamic and Testes Tissues

Cytoscape was used to analyze the protein network of the differentially expressed genes in hypothalamic and testicular tissues; association grid > five associated gene interactions were retained. There were 107 related genes and 68 upregulated genes in hypothalamic tissues (Figure 10): *PTPRC*, *C1QA*, *KDR*, *PTPN11*, *RPL8*, *RPS3A*, *RPS20*, *RPL12*, *RPS5*, *LOC101115926*, *RPS15A*, *RPL21*, *GNRH1*, *PAKCA*, *ND5*, *ND4*, *COX1*, *COX3*, *MIB1*, and *MX2* were highly correlated. In contrast, 39 downregulated genes, including *PROM1*, *BTG2*, *ITGB6*, *FOS*, *LOC101113001*, *CCDC40*, *DNAH5*, *DNAL1*, *CCDC39*, *ACTA1*, and *ACTN2*, were highly correlated.

There were 18 associated genes with >5 genes in testicular tissues. Thirteen upregulated genes (Figure 11), including *PRIM1*, *DSCC1*, *MCMDC2*, *MCM6*, *TYMS*, *GMNN*, and *JAK2*, were highly correlated. Five downregulated genes, including *FEN1*, *CDKN1A*, and *CDK6*, also showed a high correlation.

The upregulated genes, *PTPRC*, *PTPN11*, *KDR*, *CD44*, *RPL8*, *RPS20*, *RPL12*, *RPL21*, *RPS3A*, *RPS5,* and *RPS15A*, in hypothalamic tissues were highly correlated with the upregulated genes, *JAK2* and *LOC101115026,* in testicular tissues (Figure 12).

### 3.10. Combined Transcriptomic and Metabolomic Analysis

The correlation analysis between the ram hypothalamic transcriptome and serum metabolome revealed 30 common metabolic pathways between hypothalamic and serum differential metabolites (Figure 13A). Conversely, the correlation analysis between the ram testicular transcriptome and serum metabolome indicated 22 common metabolic pathways between testicular and serum differential metabolites (Figure 13B).

Notably, taurine and low taurine metabolism (ko00430); arginine and proline metabolism (ko00330); purine metabolism (ko00230); the pentose phosphate pathway (ko00030); cysteine and methionine metabolism (ko00270); ABC transport (ko02010); valine, leucine, and isoleucine degradation (ko00280); fructose and mannose metabolism (ko00051); 2-oxycarboxylic acid metabolism (ko01210); and tryptophan metabolism (ko00380) were significant in hypothalamic tissues (Figure 13C,D).

Furthermore, amino acid biosynthesis (ko01230); nitrogen metabolism (ko00910); glutamatergic synapse (ko04724); glycine, serine, and threonine metabolism (ko00260); arginine biosynthesis (ko00220); glutathione metabolism (ko00480); cysteine and methionine metabolism (ko00270); fructose and mannose metabolism (ko00051); proximal tubule bicarbonate reclamation (ko04964); and purine metabolism (ko00230) were significant in testicular tissues (Figure 13E,F).

According to the transcriptomic and metabolomic correlation analysis, *L*-Cit was possibly absorbed by the intestine and metabolized in the liver before accessing blood metabolism; conversely, amino acid metabolism in the blood increased significantly to improve the amino acid levels in seminal plasma. The testes possibly increase the pyruvate content of seminal plasma via glycolysis/gluconeogenesis, which provides energy to sperm; however, the sugar content in seminal plasma was decreased. Additionally, amino acid biosynthesis and nitrogen metabolism in the blood possibly promote the increase in ribosomal protein genes in the hypothalamus and testes, consequently improving the semen quality of rams (Figure 13G).

## 4. Discussion

The present study performed the targeted metabolic analysis of ram seminal plasma after *L*-Cit supplementation and showed increased contents of most amino acids as well as decreased sugar contents, except for xylose. Additionally, following the nontargeted ram serum analysis, protein digestion and absorption, mineral absorption, and aminoacyl tRNA biosynthesis were observed to significantly increase. Overall, 15,256 and 18,829 genes were detected in ram hypothalamic and testicular tissues, respectively. After feeding with *L*-Cit, 424 and 700 genes were specifically expressed in hypothalamic and testicular tissues, respectively. Following the KEGG analysis, protein digestion and absorption, in addition to the interaction of the neuroactive ligand–receptor pathways, were highly significant in ram hypothalamic tissues. Furthermore, disease-related pathways, glycolysis/gluconeogenesis, amino acid and nucleotide sugar metabolism, galactose metabolism, and the HIF-1 signaling pathway were highly significant in testicular tissues.

In this study, we investigated the nontargeted metabolism of ram plasma from 0 to 72 days after *L*-Cit supplementation. The results showed that pyrimidine and phenylalanine metabolism, in addition to the arginine biosynthesis pathway, were the most significant pathways, indicating their crucial role in the reproductive performance of rams. The experiment began in early August, and continued till mid-November. This period was selected because it is crucial for the transition from the nonbreeding to the breeding season of rams. Our findings indicate that the abovementioned three pathways are important for the regulation of this transition. However, transcriptomic and serum association analyses of hypothalamic tissues indicated an increased expression of purine metabolism and metabolism-related genes of taurine and low taurine. Furthermore, pyrimidine nucleotides are vital in synthesizing DNA, RNA, glycoproteins, and phospholipids, including participating in cell metabolism and proliferation [16]. Two possible reasons for the accelerated metabolism of pyrimidines in the organism are the promotion of sperm production by five consecutive days of semen collection and the accelerated transition of the nitrogen cycle from the non-breeding to the breeding phase.

Studies have shown that *L*-Cit does not degrade in the rumen of sheep and cattle [13,14]. After entering the body, *L*-Cit generates arginine under the action of argininosuccinate synthase and lyase [17]. *N*-acetyl-*L*-glutamate is synthesized by acetyl CoA and glutamate, mediated by *N*-acetylglutamate kinase, which catalyzes arginine biosynthesis. Notably, *N*-acetyl-*L*-glutamate is an allosteric catalyst for carbamoyl phosphate synthase in mammals [18]. In addition, citrulline synthesis from carbamoyl phosphate and ornithine is catalyzed by carbamoyl phosphate synthase [19]. Therefore, *N*-acetyl-*L*-glutamate is a key cofactor of the carbamoyl phosphate synthase urea cycle. Arginine hydrolysis produces urea and NO in the body [20], consequently contributing to the arginine and NO pathways. As such, *L*-Cit exerts a synergistic role with *L*-Arg and *L*-ornithine. Studies have proven that *L*-Cit may participate in protein synthesis and metabolism [21]. Therefore, *L*-Cit can improve the nitrogen balance of the body more effectively than *L*-Arg and other nonessential amino acids [22]. Moreover, *L*-Cit can improve muscle protein synthesis under dietary protein limitation conditions [23]. After *L*-Cit supplementation in the present study, *L*-valine, *L*-threonine, *L*-arginine, *L*-tryptophan, *L*-isoleucine, *L*-lysine, *L*-methionine, *L*-leucine, *L*-aspartic acid, acetyllysine, sarcosine, *L*-homoserine, and *N, N*-dimethylglycine serum levels were increased significantly, including the levels of *L*-proline, *L*-citrulline, *L*-glutamate, *L*-arginine, *L*-pyroglutamic acid, and *L*-*O*-phosphoserine. The protein digestion and absorption pathways were significant in the equivalent nontargeted metabolism and transcriptomic analyses, demonstrating that *L*-Cit promotes amino acid synthesis and metabolism.

*L*-Cit is mainly absorbed by the kidney and participates in the urea cycle to maintain blood ammonia balance. Under normal physiological conditions, the blood ammonia concentration in the body is relatively low. However, after the obstacle of urea synthesis, the blood ammonia concentration increased, thereby increasing the concentration of ammonia in the brain, decreasing α-ketoglutarate levels, and weakening the tricarboxylic acid cycle, which consequently reduces energy metabolism while increasing glutamate and glutamine levels. In this study, *L*-Cit metabolism may increase the body’s blood ammonia concentration and at the same time improve the body’s energy metabolism in order to eliminate urea disorders. Studies have demonstrated that glucose metabolism is inhibited by the blockage of glutamine metabolism in the body [24]. Following *L*-Cit feeding, glucose levels in seminal plasma did not exhibit any significant changes; however, over time, they decreased, which may be attributed to continuous semen collection. Gonzales et al. reported that the fructose concentration in seminal plasma did not significantly affect the sperm index [25], and Elzannty et al. revealed that it had no significant effect on sperm motility [26]. There was no significant difference in fructose content between the semen of *L*-Cit-supplemented rams and that of the control group; however, it decreased significantly over time. This may be because the fructose content in ram semen decreased due to the weekly semen collection. Studies have reported that in in vitro bacterial cultures the pentose phosphate pathway is a major pathway of xylose metabolism, and glucose exhibited an inhibitory effect on the glycolysis metabolites used by xylose [27,28]. In this experiment, *L*-Cit supplementation significantly increased the xylose content in semen over time. However, this may be due to the direct decomposition of glucose, and the inhibition of xylose utilization may be related to the metabolic pathway of *L*-Cit in rams. Additionally, in the targeted metabolism of seminal plasma, the contents of most sugars tended to decrease, except for those of xylose and pyruvate, which increased significantly. This can be related to glycolysis/gluconeogenesis, amino acid and nucleotide sugar metabolism, and galactose metabolism. Furthermore, the correlation analyses of the testicular tissue transcriptome and serum metabolome showed increased expression of fructose and mannose metabolism genes, indicating that *L*-Cit can promote glucose metabolism and energy by increasing the pyruvate content in seminal plasma.

In this study, the GO enrichment analysis of differential genes showed that, following *L*-Cit supplementation, the G protein-coupled receptor (GPCR) signal pathway, cell movement or the movement of subcellular components and microtubule-related complexes, and cell components in ram hypothalamic tissues were significant. The GPCR pathway affects cell proliferation and differentiation. In the body, estrogen has rapid non-classical genomic effects independent of the estrogen receptors α and β, and this process is mediated by estrogen receptor GPCR 30 molecules in the body’s cell membrane [29], which affect the proliferation and differentiation of cells in the body. Additionally, on the binding of estrogen to GPCR 30, a rapid physiological response via the GPCR 30 molecule/epidermal growth factor receptor pathway is produced [30], affecting cell proliferation and differentiation. Knocking out the GPCR 30 gene in mice hindered follicle maturation [31]. The mitogen-activated protein kinase is the main enzyme in the GPCR pathway in mammals, which mediates the physiological function of gonadotropin-releasing hormone in vivo [32]. Additionally, the mitogen-activated protein kinase depends on connexin 43. The phosphorylation of the mitogen-activated protein kinase can reduce the gap permeability of the goat oocyte coronoid colliculus complex and regulate goat oocyte meiosis [33]. The movement of cellular or subcellular components requires the Wnt signaling pathway, which involves numerous physiological processes. *PYGO2* can regulate cell differentiation, gene transcription, and spermatogenesis through Wnt-dependent mechanisms [34]. However, kinesin conformation can transform chemical energy into mechanical energy and produce relative microtubule orbital movement in life activities, such as vesicle movement and cell division. In contrast, microtubule movement is critical in the body’s vesicle movement and cell division [35], showing that *L*-Cit can affect reproduction-related signal pathways in the ram hypothalamus, which consequently regulates the reproductive performance of rams.

Protein complexes are associated with glucose metabolism, the nervous system, and male sterility. In proteomic studies, protein complexes of the endoplasmic reticulum (ER) membrane participate in the degradation of ER proteins, form ER mitochondrial thrombi, and participate in the correct arrangement of multiple transmembrane proteins [36,37]. With a homozygous ER membrane protein complex 10 knockout mice were completely infertile, and the fertility of heterozygous mice decreased [38]. Supplementing *L*-Cit to rams significantly increased MHC proteins, MHC class II proteins, plasma membrane proteins, and force protein complexes in the plasma membrane as well as cell components in ram testicular tissues. Consistent with the improved ram semen quality shown previously [1], *L*-Cit regulated protein complexes in ram testicular tissues, subsequently improving semen quality.

Although minerals cannot be synthesized in the body, they are physiologically important. Notably, all minerals except iron directly or indirectly affect male animals [39,40]. Supplementation with zinc, manganese, selenium, and cadmium has been shown to improve semen quality in animals [41,42]. Conversely, calcium ions play an important role in cell growth. A very high calcium ion concentration can activate calcium-ion-sensitive protein degrading enzymes, resulting in cell decomposition [43]. If the ER calcium ion pool is depleted, it promotes the expression of apoptosis-related genes, leading to apoptosis. Concurrently, increased calcium ion concentration in the mitochondria can trigger programmed apoptosis [44]. Furthermore, our findings showed that supplementing rams with *L*-Cit improved mineral absorption in the serum as well as semen quality.

The extracellular matrix contains proteoglycans, glycoproteins, and elastic fibers. Studies have demonstrated that fibroblasts regulate the formation of the extracellular matrix in the body. The extracellular matrix not only provides physical support for tissue growth but also serves as a reservoir of cytokines. Furthermore, it regulates fibroblasts by regulating cell growth factors and surface receptors [45], as well as these cell–matrix interactions with cell surface receptors [46]. Components such as fibronectin and collagen can provide scaffolds for contact guidance and control cell adhesion as well as migration [47]. Extracellular matrix decomposition products act as chemotactic and opsonin and participate in the secretion of basic fibroblast growth and vascular endothelial factors, which are required for cell secretion [48]. Notably, on supplementing rams with *L*-Cit in the present study, the structural components of the extracellular matrix increased significantly in hypothalamic tissue MF. Additionally, cell adhesion, biological adhesion, homophilic cell adhesion through plasma membrane adhesion molecules, intercellular adhesion, and intercellular adhesion pathways increased significantly in testicular tissue MF. Based on these results, *L*-Cit may act on the hypothalamus, thereby regulating the gonadal activity of rams.

The neuroactive ligand–receptor interaction signal pathway includes all of the receptors and ligands associated with intracellular and extracellular signal pathways on the ER membrane. However, in this pathway a biogenic amine is vital in the body’s physiological and endocrine rhythm [49]. In this study, the regular evaluation of ram semen quality indicated that physiological and endocrine rhythm significantly impacted semen collection. After supplementation with *L*-Cit, the interaction signal pathway of the neuroactive ligand–receptor increased significantly, showing that *L*-Cit can regulate the physiological and endocrine rhythm of rams, subsequently regulating their reproductive performance. Studies have shown that the HIF-1 signaling pathway affects the body’s antioxidant performance and glucose metabolism. Notably, *Nrf2* in the HIF-1 signaling pathway maintains the body’s mitochondrial physiological function, glucose metabolism and synthesis, cell redox, and normal protein function [50]. Moreover, the correlation analysis between the testicular tissue transcriptome and serum metabolomics in this experiment showed increased glutathione metabolism and increased fructose- as well as mannose-metabolism-related gene expression, indicating that *L*-Cit is involved in antioxidative performance and glucose metabolism, which regulate ram semen quality.

ABC transport is broadly associated with various physiological functions in the body [51]. Notably, aminoacyl tRNA regulates RNA replication, translation, and protein interactions, which produce, transport, and store sperm [52]. This may be related to the enhancement of sperm density, sperm motility, number of rectilinear sperm, and sperm mitochondrial membrane potential by *L*-Cit in rams

## 5. Conclusions

Supplementing rams with *L*-Cit regulated the protein digestion and absorption pathway, improved the serum–amino acid metabolism pathway, regulated testicular-related genes, increased the testicular glycolysis/gluconeogenesis pathway, reduced the sugar contents in seminal plasma, and increased amino acid levels in seminal plasma. All of these effects resulted in improved ram semen quality.

## Figures and Tables

**Figure 1 animals-13-00217-f001:**
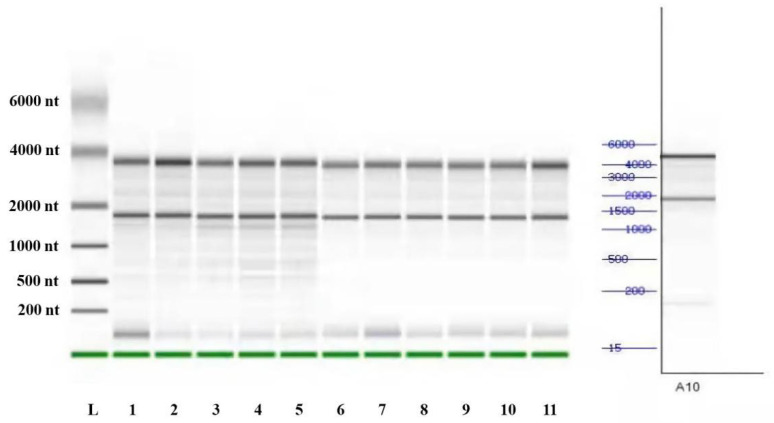
RNA results.

**Figure 2 animals-13-00217-f002:**
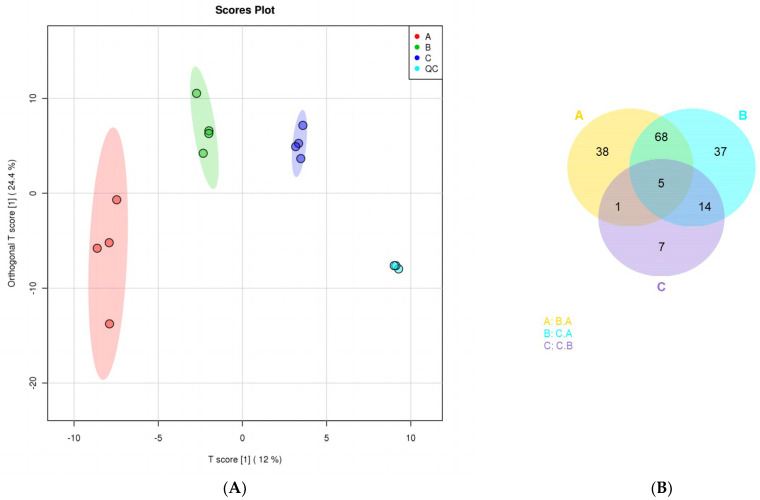
Hierarchical cluster analysis thermodynamic diagram of each comparison. (**A**) The serum sample for day 0; (**B**) control group serum sample for day 72; (**C**) test group serum sample for day 72.

**Figure 3 animals-13-00217-f003:**
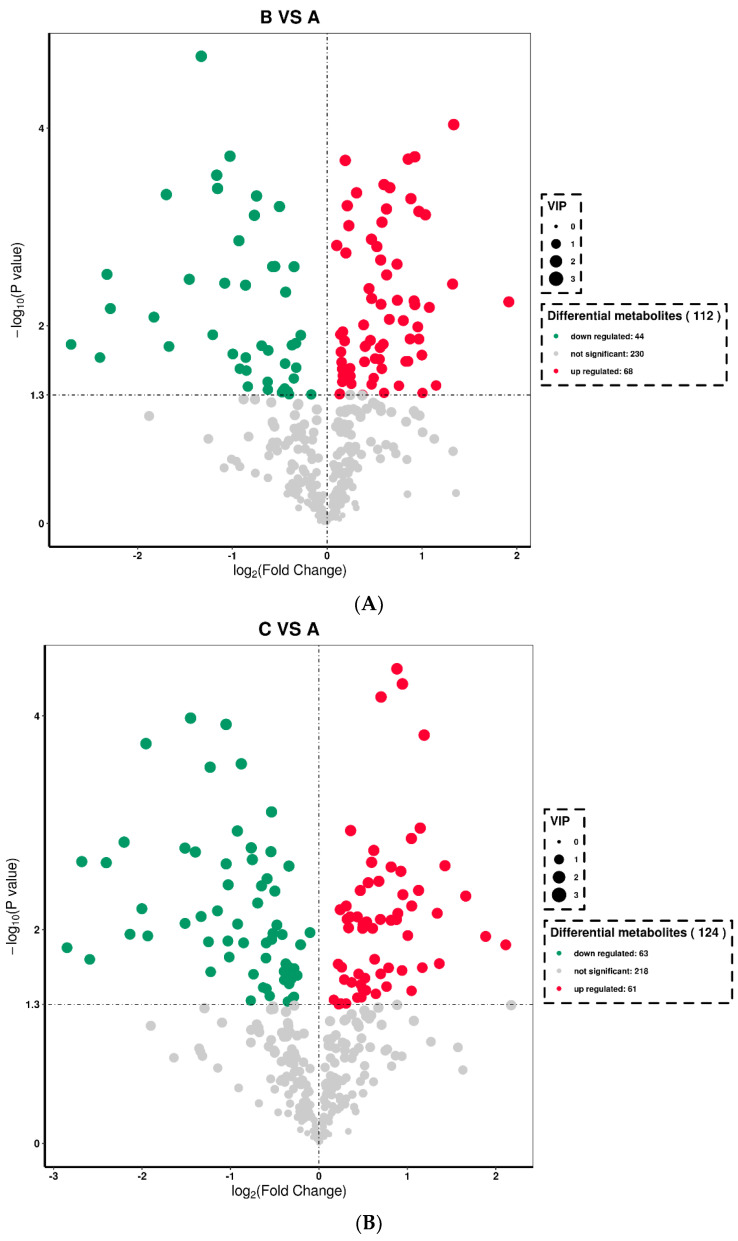
Volcano plots of differential metabolites (**A**−**C**.). Each point in the volcanic map indicates a metabolite, the abscissa represents multiple changes in each substance compared in this group (log base 2), and the ordinate represents the *p*-value of the t-test (log base 10). The scatter color represents the final screening result. Metabolites with significant differences are shown in red and green, whereas those without significant differences are shown in gray. (**A**) Ram serum before *L*-Cit feeding; (**B**) ram serum in the control group on the day 72; and (**C**) ram serum in the test group on day 72.

**Figure 4 animals-13-00217-f004:**
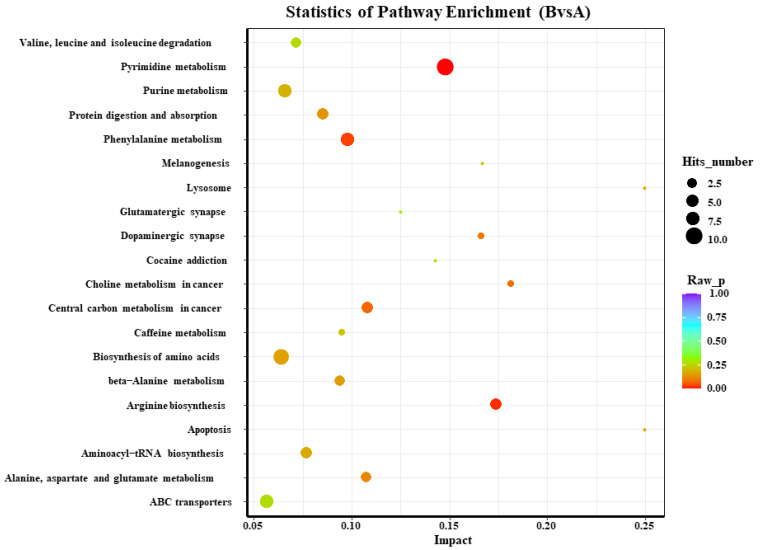
KEGG enrichment analysis of serum metabolism on days 0 and 72 of the control group.

**Figure 5 animals-13-00217-f005:**
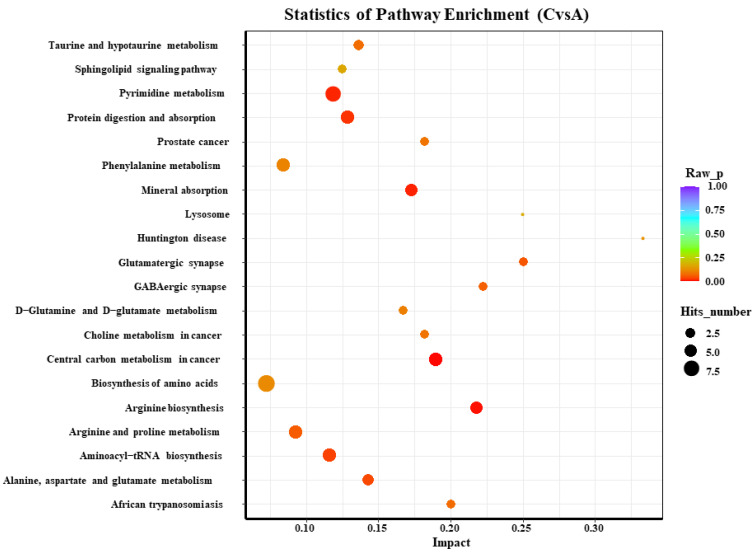
KEGG enrichment analysis of serum metabolism on days 0 and 72 of the *L*-Cit-supplemented group.

**Figure 6 animals-13-00217-f006:**
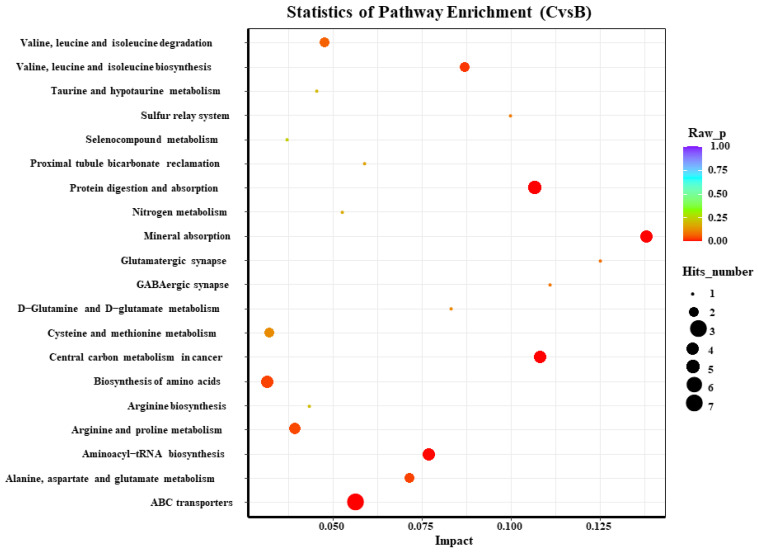
KEGG enrichment analysis of the serum metabolism on day 72 by comparing the control and *L*-Cit-supplemented groups.

**Figure 7 animals-13-00217-f007:**
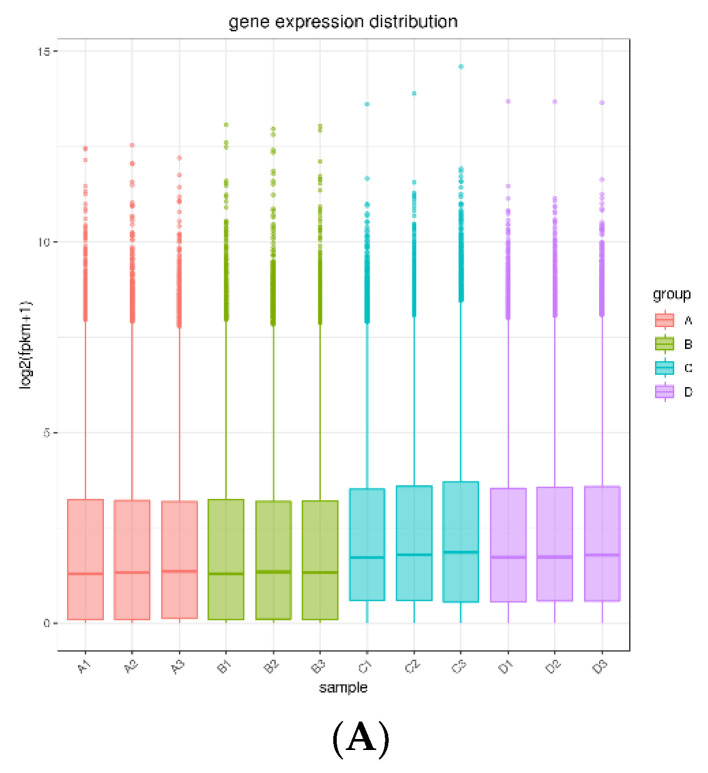
(**A**): Quality statistics of hypothalamic and testicular tissue sequencing data. (**B**,**C**): Hierarchical cluster analysis thermodynamic diagram of hypothalamic and testicular tissue.The abscissa contains the sample name; the ordinate is log2 (fpkm + 1). (**D**–**F**): Venn diagram. (**G**,**H**): Differential gene volcano map of hypothalamus and testis of rams. (**A**) Hypothalamic tissue of the control group; (**B**) hypothalamic tissue of the test group; (**C**) testicular tissue of the control group; and (**D**) testicular tissue of the control group.

**Figure 8 animals-13-00217-f008:**
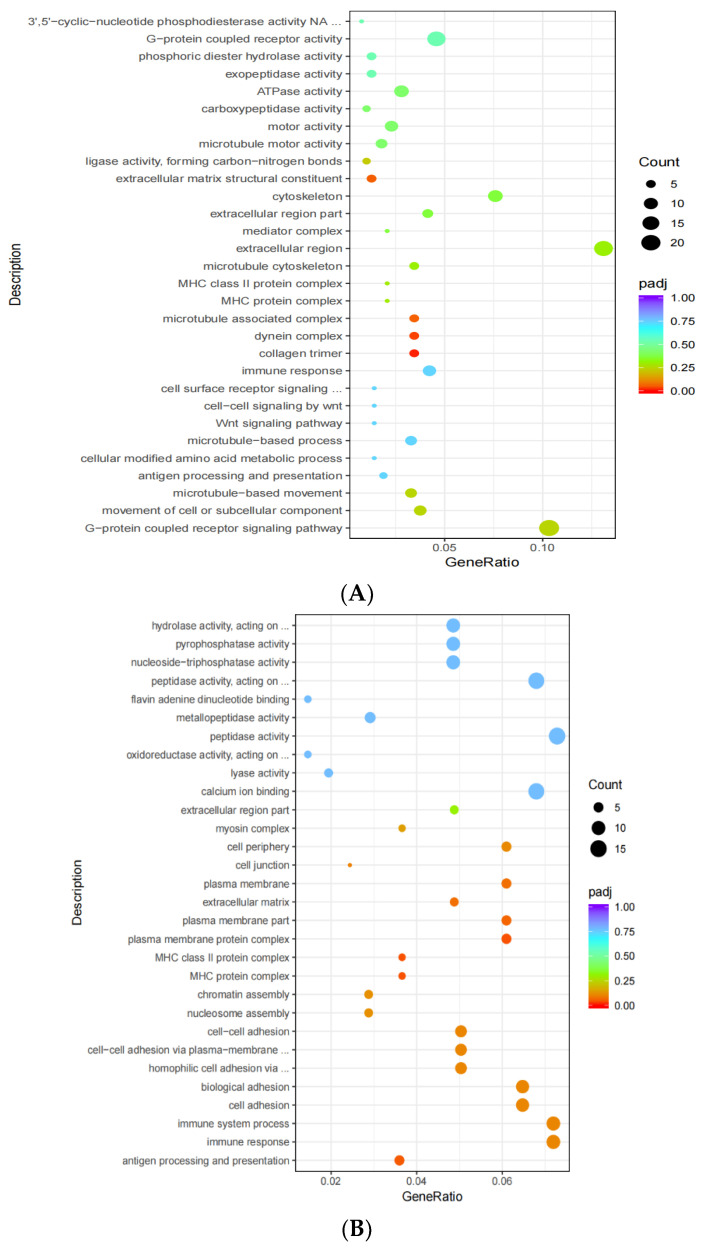
GO enrichment analysis of differentially expressed genes by comparing tissues of the control (**A**) and *L*-Cit-supplemented group (**B**).

**Figure 9 animals-13-00217-f009:**
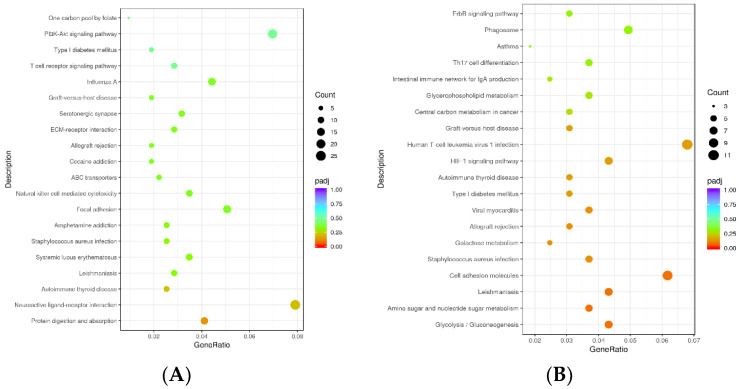
Scatter diagram of KEGG enrichment in hypothalamic (**A**) and testicular (**B**) tissues.

**Figure 10 animals-13-00217-f010:**
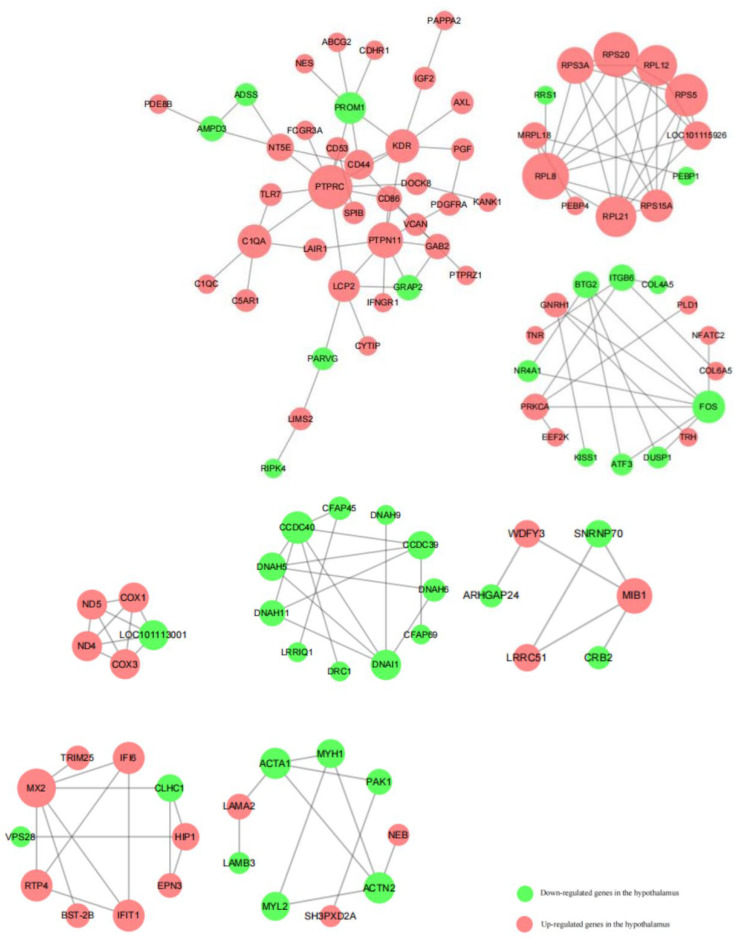
Protein interaction network of differentially expressed genes by comparing the hypothalamic tissues of the control and *L*-Cit-supplemented groups. Red indicates upregulated genes, green indicates downregulated genes, and larger, wired images indicate higher intergenic associations.

**Figure 11 animals-13-00217-f011:**
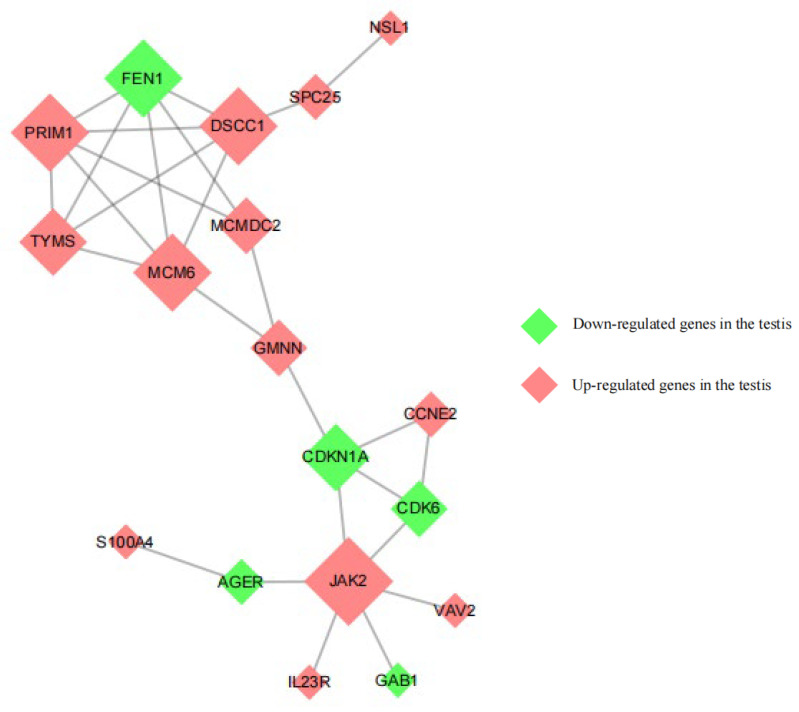
Protein interaction network in ram testicular tissues.

**Figure 12 animals-13-00217-f012:**
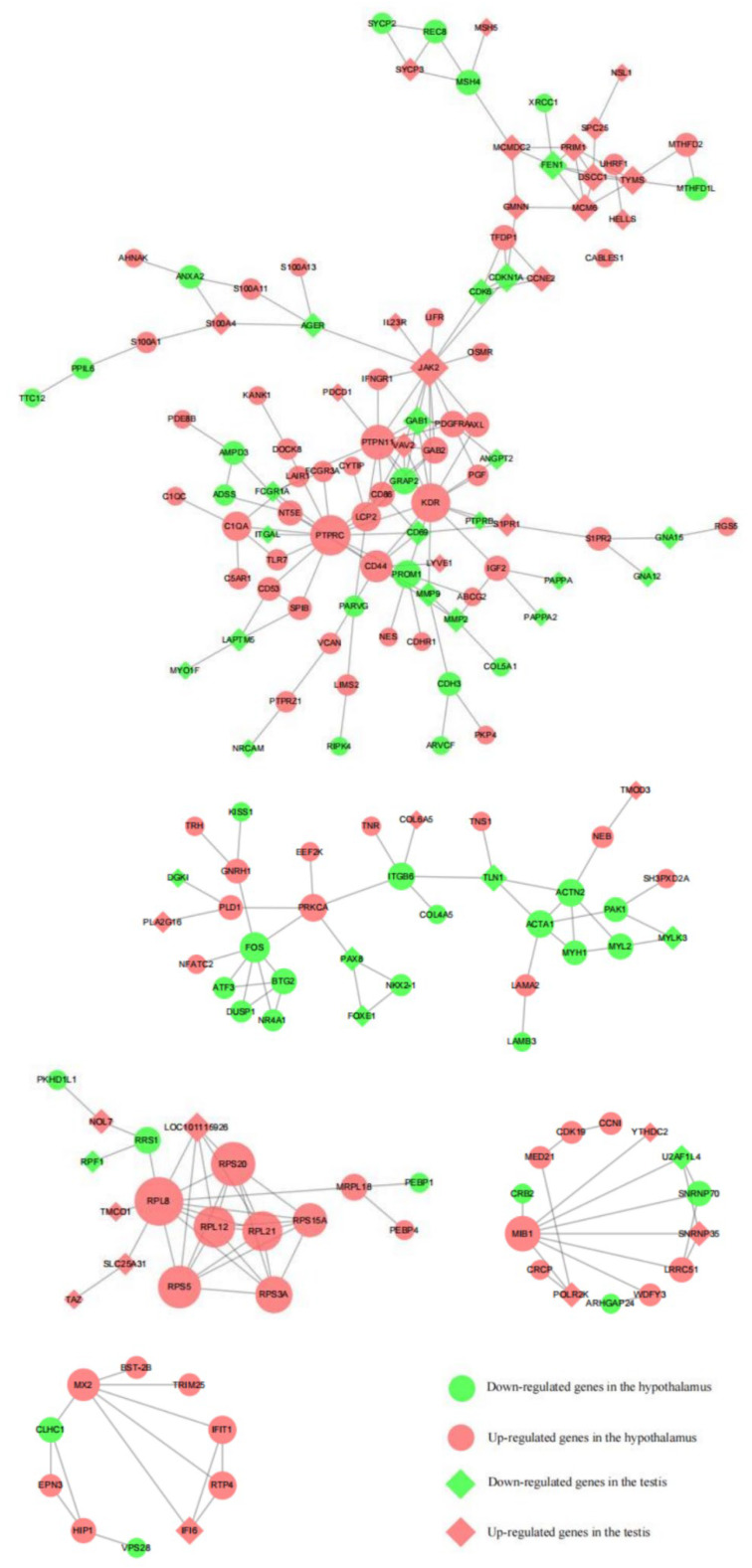
Protein interaction network in ram hypothalamic and testicular tissues.

**Figure 13 animals-13-00217-f013:**
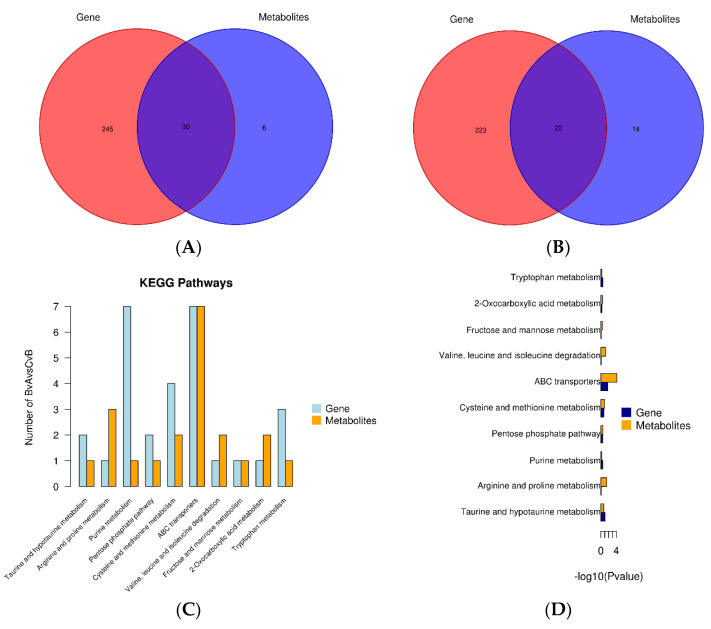
Landscape of the metabolic pathway and key genes in the semen, serum, hypothalamus, and testes of rams after *L*-Cit supplementation from our metabolome–transcriptome analyses and review of other research.

**Table 1 animals-13-00217-t001:** Diet compositions and nutritional levels of rams (based on dry matter, %).

Diet Composition	Proportion	Nutritional Level	Content
Alfalfa hay	25.25	Organic matter	90.38
Wheat straw hay	32.61	CP	10.41
Hay	27.70	NDF	47.91
Concentrate supplements	12.21	ADF	31.87
Baking soda	1.23	Ca	1.31
NaCl	1.00	P	0.19
Total	100.00		

Note: The crude protein (CP), neutral detergent fiber (NDF), acid detergent fiber (ADF), calcium (Ca), phosphorus (P), and nutritional ingredient amounts are actual measured values.

**Table 2 animals-13-00217-t002:** Statistics.

Mode	Features	Metabolites
Pos	19948	224
Neg	14525	180

**Table 3 animals-13-00217-t003:** Effects of *L*-Cit supplementation on ram seminal plasma (*n* = 4, ug/mL).

Items	Control Group	Test Group	SME	*p*		
				trt	Date	t × d
Glucose	58.78	56.45	2.39	0.5116	0.0030	0.5116
Fructose	41.72	40.29	2.82	0.7291	<0.0001	0.7291
Xylose	5.54	7.97	0.22	<0.0001	<0.0001	<0.0001
Rhamnose	0.26	0.25	0.03	0.7442	0.6906	0.7442
Arabinose	24.57	22.17	0.47	0.0066	<0.0001	0.0066
Maltose	90.12	81.50	1.81	0.0099	<0.0001	0.0099
Ribose	20.79	16.23	1.05	0.0156	<0.0001	0.0156
Lactose	0.47	0.59	0.05	0.1295	0.1595	0.1295
Xyloketose	19.10	21.13	0.67	0.0655	0.0057	0.0655
Pyruvate	63.77	167.53	3.67	<0.0001	<0.0001	<0.0001

Note: Winning bids with different lowercase letters for the same index indicate significant differences (*p* < 0.05); markings with the same letter and no markings indicate that the difference is not significant (*p* > 0.05); and winning bids for the same index with different capital letters indicate that the difference is extremely significant (*p* < 0.01).

**Table 4 animals-13-00217-t004:** Effects of *L*-Cit supplementation on amino acid levels in ram seminal plasma (*n* = 4, ng/mL).

Items	Control Group	Test Group	SME	*p*		
				trt	Date	t × d
*L*-citrulline	123.42	140.35	4.16	0.0194	0.0289	0.0194
*L*-arginine	276.91	379.95	18.16	0.0039	0.0042	0.0039
*L*-valine	440.17	599.49	4.58	<0.0001	<0.0001	<0.0001
*L*-threonine	109.43	162.19	4.13	<0.0001	<0.0001	<0.0001
*L*-lysine	7103.80	9285.13	429.28	0.0071	0.0036	0.0071
*L*-histidine	1517.67	1522.03	56.03	0.9575	0.5056	0.9575
*L*-tryptophan	189.20	393.81	21.44	0.0001	0.0013	0.0001
*L*-leucine	634.70	713.96	13.10	0.0027	0.2871	0.0027
*L*-phenylalanine	212.54	181.67	10.48	0.0709	0.0002	0.0709
*L*-isoleucine	669.82	757.03	4.07	<0.0001	<0.0001	<0.0001
*L*-methionine	37.23	78.09	3.83	<0.0001	0.0004	<0.0001
*L*-serine	1103.36	1144.57	19.22	0.1680	0.6839	0.1680
*L*-pyroglutamic acid	35.97	44.96	2.02	0.0135	0.0636	0.0135
*N*-acetyl-*L*-alanine	2.08	2.85	0.37	0.1827	0.0001	0.1827
*L*-alanine	625.29	707.02	36.93	0.1562	0.0039	0.1562
Dimethylglycine	45.02	61.39	2.49	0.0017	0.0126	0.0017
*L*-proline	1456.15	2149.2	148.46	0.0108	0.0188	0.0108
*L*-pipecolic acid	4.61	4.67	0.33	0.8978	0.9751	0.8978
*L*-hydroxyproline	16.81	18.77	1.44	0.3674	0.7843	0.3674
*L*-ornithine	1133.81	1227.77	19.90	0.0102	0.0010	0.0102
L-aspartate	3138.39	3962.72	89.23	0.0002	<0.0001	0.0002
1-methyl-histidine	23.92	27.60	1.62	0.1466	0.1252	0.1466
*D*-tyrosine	291.40	277.25	18.70	0.6072	0.4956	0.6072
Acetyllysine	2.13	2.62	0.09	0.0055	0.0040	0.0055
*N*-acetyl-glutamate	75.36	76.63	1.45	0.5539	0.6039	0.5539
*L*-kynurenine	13.88	14.86	0.42	0.1367	<0.0001	0.1367
*L*-cystine	33.73	40.91	3.16	0.1460	0.0078	0.1460
Sarcosine	605.74	693.30	11.95	0.0008	<0.0001	0.0008
*L*-hydroxyisocaproic acid	3.99	5.10	0.24	0.0110	0.0518	0.0110
*S*-adenosyl-*L*-homocysteine	8.23	9.70	1.60	0.5365	0.8238	0.5365
Glycine	0.56	0.43	0.04	0.0427	0.0233	0.0427
*L*-homoserine	13.98	20.79	0.91	0.0008	<0.0001	0.0008
*L*-glutamate	77.17	89.90	3.30	0.0258	0.0130	0.0258
*L*-*O*-phospho-serine	2.68	3.54	0.23	0.0276	0.1586	0.0276
*S*-adenosyl-*L*-methionine	12.50	8.24	0.36	<0.0001	<0.0002	<0.0001
Homocysteine	1.84	2.50	0.36	0.2120	0.0906	0.2120

**Table 5 animals-13-00217-t005:** Summary of RNA-Seq data after quality control.

Sample	Total Reads	GC Content	Total Map	Unique Map	Multi Map
A1	48,048,384	51.09	46,290,406 (96.34%)	42,309,500 (88.06%)	3,980,906 (8.29%)
A2	47,072,662	50.87	45,353,328 (96.35%)	40,889,648 (86.86%)	4,463,680 (9.48%)
A3	45,411,406	49.42	43,788,313 (96.43%)	39,717,073 (87.46%)	4,071,240 (8.97%)
B1	45,697,300	51.33	44,069,862 (96.44%)	39,713,158 (86.9%)	4,356,704 (9.53%)
B2	46,875,500	48.75	44,994,231 (95.99%)	38,665,455 (82.49%)	6,328,776 (13.5%)
B3	45,786,310	49.72	44,033,024 (96.17%)	38,372,276 (83.81%)	5,660,748 (12.36%)
C1	45,953,970	51.95	44,523,628 (96.89%)	42,617,822 (92.74%)	1,905,806 (4.15%)
C2	45,188,818	52.93	43,742,967 (96.8%)	41,956,321 (92.85%)	1,786,646 (3.95%)
C3	44,932,746	53.52	43,541,577 (96.9%)	41,890,893 (93.23%)	1,650,684 (3.67%)
D1	51,127,152	52.57	49,558,072 (96.93%)	47,470,625 (92.85%)	2,087,447 (4.08%)
D2	47,864,170	51.88	46,383,670 (96.91%)	44,523,956 (93.02%)	1,859,714 (3.89%)
D3	45,621,730	51.97	44,250,479 (96.99%)	42,307,892 (92.74%)	1,942,587 (4.26%)

A, hypothalamic tissue of the control group; B, hypothalamic tissue of the test group; C, testicular tissue of the control group; and D, testicular tissue of the control group.

## Data Availability

The data that support the findings of this study are available from GSA: CRA006955, which are publicly accessible at https://ngdc.cncb.ac.cn/gsub/ (accessed on 19 May 2022). Other data that support the findings of this study are available from the corresponding author upon reasonable request.

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
