# Peer review of "Metabolomic and Transcriptomic Changes Underlying the Effects of L-Citrulline Supplementation on Ram Semen Quality"

_animals, 2023, doi:10.3390/ani13020217_

Round 1

Reviewer 1 Report

Manuscript ID – Animals 2053878

To the Editor and the Authors.

This manuscript is an original article about the supplementation of L Cit in the ram feed. The results obtained showed that this supplementation improves the content of amino acids in ram semen, by promoting the protein metabolism of the hypothalamus testis axis.

The manuscript is overall well written. English is fine and the sections are well structured.

All procedures and results are clearly described and interpreted; also, statistical analysis is well conducted.

My only suggestion is to revise and correct the ethical statements (lines 112-113.)

Author Response

Dear,Reviewer

We greatly appreciate the hard work, fair and constructive comments from you and reviewers. The comments are necessary for improving the whole quality of our manuscript.

Reviewer 2 Report

It is a great paper, very exhaustive and complete.

Perhaps some sections could be summarized to make it easier to read.

Thus, the material and methods section can be summarized, giving references to analytical methods, without making such an exhaustive description.

There are very interesting schemes, the scheme in figure G gives a magnificent image of the experiment.

Author Response

(The authors gave the same response as above.)

Reviewer 3 Report

The manuscript is fairly well written but there are some sections that are a bit confusing and therefore need to be rephrased. 

Title: I suggest modifying the title to “Metabolomic and transcriptomic changes underlying the effects of L-citrulline supplementation on ram semen quality”

Line 14: Delete “rams”

Line 16: “feeding 12 g/d L-Cit in rams feed”…Delete “in rams feed”

Line 18: Replace “metabonomics” with “metabolomics”

Line 22: Replace “can improved” with “can improve”

Line 27: Please mention when the samples were collected 

Lines 39-42: This statement needs to be rephrased for clarity.

Line 52-53: “Seminal plasma is an essential component of sperms and is mainly secreted by the testis, 52 epididymis, and accessory gonads.” is an odd statement. “Seminal plasma is an essential component of semen…the other component is spermatozoa. Seminal plasma is mainly secreted by the accessory sex glands. Please correct it.

Line 60-64: Add reference for this statement.

Line 108: “artificial” instead of “fake”

Line 113: HERE NAME OF ETHICAL BOARD?

Line 128-129: “Semen was collected between 1 (recorded as day 0) and 71 days before the test and morning feeding.” The authors must clearly mention the days of collection relative to the period of supplementation.

Line 130-131: “Next, two sperm samples from the same ram group were mixed and centrifuged 130 at 3500 rpm for 15 min.” This one also needs to be clarified. 

Line 617-621: That is a very long and confusing statement. Please rephrase to make it more clear.

Author Response

Title: I suggest modifying the title to “Metabolomic and transcriptomic changes underlying the effects of L-citrulline supplementation on ram semen quality”

Response: Thanks for the positive comments, and the manuscript had been revised according to the value suggestion.

Line 14: Delete “rams”

Line 16: “feeding 12 g/d L-Cit in rams feed”…Delete “in rams feed”

Line 18: Replace “metabonomics” with “metabolomics”

Line 22: Replace “can improved” with “can improve”

Response: Thanks for the positive comments, and the manuscript had been revised according to the value suggestion.

Line 27: Please mention when the samples were collected 

Response: First sentence in Highlight had been rewritten. Please see the words with yellow background in Highlight.

Lines 39-42: This statement needs to be rephrased for clarity.

Response: First sentence in Highlight had been rewritten. Please see the words with yellow background in Highlight.

Line 52-53: “Seminal plasma is an essential component of sperms and is mainly secreted by the testis, 52 epididymis, and accessory gonads.” is an odd statement. “Seminal plasma is an essential component of semen…the other component is spermatozoa. Seminal plasma is mainly secreted by the accessory sex glands. Please correct it.

Response: First sentence in Highlight had been rewritten. Please see the words with yellow background in Highlight.

Line 108: “artificial” instead of “fake”

Line 113: HERE NAME OF ETHICAL BOARD?

Response: Thanks for the positive comments, and the manuscript had been revised according to the value suggestion.

Line 128-129: “Semen was collected between 1 (recorded as day 0) and 71 days before the test and morning feeding.” The authors must clearly mention the days of collection relative to the period of supplementation.

Response: Thanks for the positive comments, and the manuscript had been revised according to the value suggestion.

Line 130-131: “Next, two sperm samples from the same ram group were mixed and centrifuged 130 at 3500 rpm for 15 min.” This one also needs to be clarified. 

Response: Thanks for the positive comments, and the manuscript had been revised according to the value suggestion.

Line 617-621: That is a very long and confusing statement. Please rephrase to make it more clear.

Response: First sentence in Highlight had been rewritten. Please see the words with yellow background in Highlight.

Reviewer 4 Report

The MS written by Zhao et al applied targeted/nontargeted metabolic (ram seminal plasma and Serum) and transcriptomic (hypothalamus testis) methods to investigate the mechanism of L-Cit supplementation on ram semen quality. The content is rich, but the writing needs to improve.

When the author compares the differential metabolites or genes, how about the foldchange? what’s the size of sample (n=? )for each group

The differential gene expression should be confirmed by real time PCR.

The writing should be more precise, methods such as the preparation of sample (serum, seminal plasma) can be simpler, and references could be cited.

The primary result of the sample identification should not be present in the materials part such as Figure 1

The discussion should not repeat the introduction or the results, such as L622, L635-639, and the pathway need more discussion

Figures should be clearer and legends should be more detailed, and titles for the table and figure should be more accurate.

There are some grammatical errors about language editing:

L15:rams semen density, ram semen density  

L19:could increased, could increase

……

L23:  hypothalamus testis axis, not accurate

L48:  reproduction-related hormones, reproductive hormones?

L127: 2.5 determination?  This title is not appropriate, Determinate what?

L131: 3500 rpm   x g

L153: too much repeat, the author may tell the difference for the detection

L265-269:  In the test group, 163 and 73 differential metabolites were annotated on days 0 and 72, not clear

Table 3. no data for specific date

Author Response

Dear,Reviewer

We greatly appreciate the hard work, fair and constructive comments from you and reviewers. The comments are necessary for improving the whole quality of our manuscript. The revision was carried out based on the point-by-point way. We have made a revision for our manuscript according to the comments, and the responses are as follows:

When the author compares the differential metabolites or genes, how about the foldchange? what’s the size of sample (n=? )for each group

Response: Thanks for the positive comments, and the manuscript had been revised according to the value suggestion.

The differential gene expression should be confirmed by real time PCR。

Response:Thanks for the positive comments, due to the epidemic situation, The experiment cannot be carried out for the time being, please forgive me.

The writing should be more precise, methods such as the preparation of sample (serum, seminal plasma) can be simpler, and references could be cited.

Response:Thanks for the positive comments, We will improve in the future.

The primary result of the sample identification should not be present in the materials part such as Figure 1

Response:Thanks for the positive comments, We will improve in the future.

The discussion should not repeat the introduction or the results, such as L622, L635-639, and the pathway need more discussion

Response: Thanks for the positive comments, and the manuscript had been revised according to the value suggestion.

  There are some grammatical errors about language editing:

L15:rams semen density, ram semen density  

Thanks for the positive comments, and the manuscript had been revised according to the value suggestion.

L19:could increased, could increase

Thanks for the positive comments, and the manuscript had been revised according to the value suggestion.

L23:  hypothalamus testis axis, not accurate

Thanks for the positive comments, and the manuscript had been revised according to the value suggestion.

L48:  reproduction-related hormones, reproductive hormones?

Thanks for the positive comments, and the manuscript had been revised according to the value suggestion.

L127: 2.5 determination?  This title is not appropriate, Determinate what?

Thanks for the positive comments, and the manuscript had been revised according to the value suggestion.

L131: 3500 rpm   x g

Thanks for the positive comments, and the manuscript had been revised according to the value suggestion.

L153: too much repeat, the author may tell the difference for the detection

Thanks for the positive comments, and the manuscript had been revised according to the value suggestion.

L265-269:  In the test group, 163 and 73 differential metabolites were annotated on days 0 and 72, not clear

Thanks for the positive comments, and the manuscript had been revised according to the value suggestion.

Table 3. no data for specific date

Thanks for the positive comments, and the manuscript had been revised according to the value suggestion.

Response: Thanks for the positive comments, and the manuscript had been revised according to the value suggestion.

Thanks for the positive comments, and the manuscript had been revised according to the value suggestion. The substantial language had been edited.

Yours sincerely Guodong Zhao